# Comparative Transcriptome Analysis Reveals Sex-Based Differences during the Development of the Adult Parasitic Wasp *Cotesia vestalis* (Hymenoptera: Braconidae)

**DOI:** 10.3390/genes12060896

**Published:** 2021-06-10

**Authors:** Yuenan Zhou, Pei Yang, Shuang Xie, Min Shi, Jianhua Huang, Zhizhi Wang, Xuexin Chen

**Affiliations:** 1Institute of Insect Science, College of Agriculture and Biotechnology, Zhejiang University, Hangzhou 310058, China; ynzhou@zju.edu.cn (Y.Z.); 1816185@zju.edu.cn (P.Y.); 11716072@zju.edu.cn (S.X.); shimin0623@zju.edu.cn (M.S.); jhhuang@zju.edu.cn (J.H.); 2Ministry of Agriculture Key Lab of Molecular Biology of Crop Pathogens and Insect Pests, Zhejiang University, Hangzhou 310058, China; 3Zhejiang Provincial Key Lab of Biology of Crop Pathogens and Insects, Zhejiang University, Hangzhou 310058, China; 4State Key Lab of Rice Biology, Zhejiang University, Hangzhou 310058, China

**Keywords:** *Cotesia vestalis*, Iso-Seq, RNA-Seq, sex-biased gene expression, lncRNA, alternative splicing

## Abstract

The endoparasitic wasp *Cotesia vestalis* is an important biological agent for controlling the population of *Plutella xylostella*, a major pest of cruciferous crops worldwide. Though the genome of *C. vestalis* has recently been reported, molecular mechanisms associated with sexual development have not been comprehensively studied. Here, we combined PacBio Iso-Seq and Illumina RNA-Seq to perform genome-wide profiling of pharate adult and adult development of male and female *C. vestalis*. Taking advantage of Iso-Seq full-length reads, we identified 14,466 novel transcripts as well as 8770 lncRNAs, with many lncRNAs showing a sex- and stage-specific expression pattern. The differentially expressed gene (DEG) analyses showed 2125 stage-specific and 326 sex-specific expressed genes. We also found that 4819 genes showed 11,856 alternative splicing events through combining the Iso-Seq and RNA-Seq data. The results of comparative analyses showed that most genes were alternatively spliced across developmental stages, and alternative splicing (AS) events were more prevalent in females than in males. Furthermore, we identified six sex-determining genes in this parasitic wasp and verified their sex-specific alternative splicing profiles. Specifically, the characterization of *feminizer* and *doublesex* splicing between male and female implies a conserved regulation mechanism of sexual development in parasitic wasps.

## 1. Introduction

The genetics of sex determination in arthropods is diverse and complex, which can be either male- or female-heterogametic (with or without heteromorphic sex chromosomes), mono- or polygenic, or haplo-diploid [1,2,3]. Haplodiploidy is a widespread phenomenon in Hymenoptera, in which females are diploid and males are haploid. It can be caused by a number of different underlying molecular mechanisms. The best-understood mode of sex determination in Hymenoptera is complementary sex determination (CSD) determined by *complementary sex determination* (*csd*), which is a paralog of *feminizer* (a homolog of *transformer* in *Drosophila melanogaster*) that emerged through gene duplication and has firstly been characterized in the honeybee *Apis mellifera* [4]. Whereas in some taxa, e.g., *Nasonia vitripennis*, sex determination does not involve CSD. However, the sex-determining pathway is largely conserved among different hymenopteran [5]. Furthermore, the molecular basis of sexual development also varies greatly with many sex- or stage-specific expressed gene sets. For example, alternative splicing (AS) changes between stages, tissues, and sexes, suggesting it as a source of developmental regulation, and sexual differentiation [6,7]. It is widely known that the sex-specific alternative splicing of genes in the sex-determining pathway, i.e., *transformer* (*tra*) and *doublesex* (*dsx*), control insect sexual development. In brief, sex-specific isoforms of *dsx*, which are controlled by alternative splicing of the upstream regulator *tra*, show distinct and often opposite effects on downstream gene expression and morphological development, ultimately leading to sexual differentiation [8,9,10]. Similarly, long noncoding RNAs (lncRNAs) are emerging as key regulatory components for sexual development in animals. The roles of lncRNAs in developmental processes, such as embryogenesis and tissue differentiation, have been extensively studied in animals [11,12,13].

Parasitic wasps lay their eggs into/on the bodies of arthropod hosts that provide nutrients and other necessary elements for the development of wasp offspring. After parasitization, the wasp larvae emerge from the hosts and pupate, and they eventually eclose into adult wasps. The successful development of wasp offspring usually results in the death of their hosts; this makes the wasps candidates for biological control agents [14,15]. The endoparasitic wasp *Cotesia vestalis* (Hymenoptera: Braconidae) is an important natural enemy of the diamondback moth *Plutella xylostella* (Lepidoptera: Yponomeutidae), a global pest of cruciferous vegetables [15]. The life history of the parasitoid wasp shows that male adults are responsible for courtship and mating, while females need to search for hosts and oviposit, indicating a distinct molecular regulation strategy during the development of females and males [16,17]. Recent analysis shows that a series of olfactory genes are sex-specifically expressed between the female and male antennae of *C. vestalis* [18,19]. However, the molecular basis of sexual development and dimorphism related to behavior, physiology, and morphology in this parasitic wasp is still poorly understood.

Recently, the complete genome sequence of *C. vestalis* has been published, providing useful information for us to study the molecular basis of its sexual development [20]. To fulfill this goal, we collected samples from pharate adult and adult development of male and female wasps. Based on the high-quality genome of *C. vestalis*, as well as taking advantage of the PacBio and Illumina sequencing platforms, we obtained an informative dataset of *C. vestalis* transcripts across the pharate adult and adult development of both male and female wasps. The results provide a molecular background and potential candidate genes for future studies of the mechanisms of sexual development and differentiation.

## 2. Materials and Methods

### 2.1. Insect Rearing and Staging of Wasp Pupae

The endoparasitic wasp *C. vestalis* parasitizes the larval stage of the diamondback moth *P. xylostella* [14]. Both species were reared in an artificial climate room with 25 ± 1 °C, 65% relative humidity, and 14:10 L/D hours as previously described [21]. The different developmental stages of *C. vestalis* can be accurately observed using time and morphological characteristics. In brief, *C. vestalis* takes a total of 11 days to develop from an egg to an adult. After parasitization, the *C. vestalis* egg hatches, the wasp larva develops in the body of a host larva for seven days, and the pupal stage lasts for four days. The late second-instar wasp larva emerges from the host’s body to become a pre-pupa. As shown in Appendix A, the pre-pupa is entirely white with light red compound eyes and no obvious antennae (stage 1). On the second day (stage 2), the pre-pupa begins to molt (after spinning a silken cocoon for about 30 h), followed by showing complete dark red compound eyes, three dark red ocelli, and white antennae. Males can easily be distinguished from female pupae in this stage, as the female pupa has obvious valvula of the ovipositor and shorter antennae. On the third day (stage 3), the pupa has an entirely black thorax, black eyes, and a partially darkened abdomen. On the fourth day (stage 4), the pupa has a fully black thorax and abdomen with brown accents present on each leg. After this stage, the pupa emerges from the cocoon as a one-day-old adult. At this point, the wasps were transferred to a new container and fed with 10% honey water and maintained at 18 °C in continuous dark. Male adults usually emerge earlier than female adults.

### 2.2. Sample Collection

Because female and male pupae are easily distinguished in stage 2, we collected samples of both sexes from this stage every day until adult wasps emerged (*n* = 5). The collected samples were labeled as follows: two-day-old male pupae (PM2d), three-day-old male pupae (PM3d), four-day-old male pupae (PM4d), one-day-old male adults (AM1d), two-day-old female pupae (PF2d), three-day-old female pupae (PF3d), four-day-old female pupae (PF4d), and one-day-old female adults (AF1d). For each developmental stage, three biological replications of each sample were collected in order to check reproducibility. Collected samples were immediately frozen in liquid N_2_ and stored in a freezer at −80 °C.

### 2.3. Illumina RNA-Seq Library Construction and Sequencing

Each of the 24 samples was ground in RNA-easy Isolation reagent (Vazyme Biotech Co., Ltd, Nanjing, China) on dry ice, and then total RNA was extracted using an RNA-easy^TM^ Isolation reagent kit following the manufacturer’s protocol. The RNA integrity value (RIN) was checked by a Bioanalyzer 2100 (Agilent Technologies Co., Ltd, Palo Alto, CA, USA) as well as 1.1% agarose gel electrophoresis to assess the total RNA quality. RNA samples with RIN > 8 were used for subsequent analyses. RNA-Seq libraries were prepared with 1 μg total RNA (each sample) according to the manufacturer’s protocol (Illumina Inc., San Diego, CA, USA). Finally, we used an Illumina HiSeq 2500 platform (Illumina Inc., San Diego, CA, USA) to generate the PE (paired-end) reads.

### 2.4. Illumina Data Analysis

Raw reads with >2 N bases were filtered. The remaining reads were filtered by removing reads containing adaptors, reads containing poly-N (>10%), and low-quality reads. Quality control and reads statistics were determined by Trimmomatic (v0.35) [22] and FastQC (v0.11.8) [23]. Q20, Q30, and GC contents of the clean reads were calculated, and all downstream analyses were based on clean reads with high quality. The final clean reads were then mapped to the *C. vestalis* reference genome [20] using Hisat2 aligner with default parameters [24]. We used the reference genome of *C. vestalis* CvesOGS1.0 to guide the transcript assembly of each sample using the StringTie program (v2.1.1, with parameters: -p 26, -c 2.5 -j 5 -m 500) [25]. Then, the output GTF files were merged into a single unified transcript using the StringTie merge function (with parameters: --merge -F 1 -m 500 -p 26 -f 0.1).

### 2.5. PacBio Library Construction and Sequencing

Total RNA of the 24 samples was mixed together in equal parts (0.5 μg for each sample). Poly(A) RNA was isolated using the Poly(A) Purist TM Kit (Ambion Inc., Grand Island, NY, USA). RNA was reverse transcribed into cDNA using the SMARTer™ PCR cDNA Synthesis Kit (Clontech Laboratories Inc., Mountain View, CA, USA). We used the BluePippin size selection system (Sage Science, Beverly, MA, USA) to generate cDNA fractions with different sizes, including 1–2 kb, 2–3 kb, and 3–6 kb. These libraries were then constructed with the Pacific Biosciences’ SMRTbell Template Prep Kit 2.0 (Pacific Biosciences, https://www.pacb.com, accessed on 1 August 2019) according to the manufacturer’s protocol. A total of five SMRT cells (two SMRT cells for libraries of 1–2 kb, two SMRT cells for libraries of 2–3 kb, and one SMRT cell for libraries of 3–6 kb) were sequenced on the PacBio RS II platform. The raw sequencing reads from the PacBio RSII SMRT cells were processed through the SMRT-Portal analysis suite to subread sequences for further processing.

### 2.6. PacBio Data Analysis

The standard protocol of Iso-Seq (Pacific Biosciences, SMRT Analysis 2.3.0, https://smrt-analysis.readthedocs.io/en/latest/SMRT-Analysis-Software-Installation-v2.3.0/, accessed on 8 March 2020) was used to process the raw PacBio full-length Iso-Seq data. After trimming the sequencing adapters, poly(A) tails, and low-quality bases, Iso-Seq reads were filtered with a minimum length of 35 bps and accuracy less than 0.75. The circular consensus sequences (CCS) were generated from the subread BAM files, also known as the reads of insert (ROI). After examining for 5′ and 3′ adaptors and poly(A) signals, full-length (FL) and non-full-length (nFL) cDNA reads were defined. Then, the chimeric reads were removed, including sequencing primers. The full-length non-chimeric (FL-NC) reads were aligned using the ICE Quiver algorithm, and then similar sequences were assigned to a cluster. Every cluster was identified as a uniform isoform. Furthermore, non-full-length cDNA reads were applied to polish each cluster. The isoform sequences with accuracy greater than 99% were considered as high-quality consensus isoforms, whereas the remaining data were defined as low-quality isoforms. These polished consensus sequences were further subjected to correction and removal of redundancy with the more accurate Illumina short reads using the LoRDEC tool [26]. Subsequently, the modified low-quality isoforms and high-quality isoforms were combined as high-quality full-length transcripts. Lastly, the redundant isoforms were removed using CD-HIT-EST [27] (v4.8.1, with the parameters: -c 0.99 -T 6 -G 1 -U 10 -s 0.999 -p 1) to yield non-redundant, non-chimeric full-length transcripts with high accuracy.

### 2.7. LncRNA Identification and Function Analysis

To identify potential lncRNAs in the PacBio Iso-Seq data, transcripts that were longer than 200 bp were retained. Furthermore, BLASTX against the Swiss-Prot database was conducted to discover protein-coding transcript homologs. We used getorf (http://emboss.bioinformatics.nl/cgi-bin/emboss/getorf, accessed on 1 October 2020) to discard open reading frames of more than 300 nucleotides. Additionally, we used Coding Potential Calculator (CPC, version 0.9-r2) [28] and CPAT (v1.2.4) [29] to evaluate the coding potential of transcripts. Combining these computational approaches, the predicted lncRNAs had to meet the following requirements: (1) transcripts that had ORFs larger than 300 nucleotides were deleted; (2) transcripts that were homologous to known protein-coding transcripts were discarded; (3) transcripts with CPC scores larger than 0 or CPAT scores larger than 0.39 were abandoned; (4) transcripts that were known rRNA, scRNA, snoRNA, snRNA, or tRNA were eliminated. The remaining transcripts from the intersection results were considered as non-coding. To avoid false positives in lncRNA identification, we only retained the lncRNAs that were mapped to the *C. vestalis* genome.

In order to perform functional annotation of *C. vestalis* lncRNAs, we predicated the target mRNA genes based on the *cis* and *trans* principle by searching for the protein-coding genes located ~10 kb upstream or downstream of the identified lncRNAs. All identified neighboring genes were used for the Gene Ontology (GO) and Kyoto Encyclopedia of Genes and Genomes (KEGG) enrichment analysis.

### 2.8. Analysis of Differentially Expressed Genes

The *C. vestalis* gene expression levels among the various samples were analyzed based on the short-read datasets generated by the Illumina sequencing platform. The gene expression levels were measured and normalized as FPKM (fragments per kilobase of transcript, per million fragments sequenced) by StringTie (v2.1.1) [25]. All the read counts of each sample were normalized to the mean total number of reads across all samples. Then, the differential expression analysis of each comparison combination was performed using the DESeq R package (v1.10.1) mode based on the negative binomial distribution from three biological replicates. Differential expression analyses in the different groups were performed using the R package edgeR v3.22.3, and an absolute value of |log_2_ (fold change)| > 1 and a *q*-value of < 0.05 were set as the criteria for significant differential expression. In order to obtain the expression profiles of lncRNA in different developmental stages, we used RSEM [30] to calculate the expression level for each of the 24 samples.

### 2.9. Function Annotation of Differentially Expressed Genes

GO annotations and KEGG metabolic pathway analysis were performed using OmicShare tools (www.omicshare.com/tools, accessed on 1 November 2020). For GO annotation, transcripts were classified into three main GO categories: cellular component (CC), molecular function (MF), and biological process (BP).

### 2.10. Identification of Differential Alternative Splicing Events

We combined sequencing data from both Iso-Seq and RNA-Seq to identify AS events in pharate adult and adult development of female and male *C. vestalis.* For the Iso-Seq data, we employed GMAP (v2020-10-27) [31] tools to map the high-quality PacBio transcripts to the reference genome CvesOGS1.0 with the default parameters: -f samse -z sense_force -t 26 -n 0. Then, samtools (v1.3) [32] was used to sort the sam format with default parameters. Finally, redundant transcripts were removed using collapse_isoforms_by_sam.py (Pacific Biosciences, https://www.pacb.com, accessed on 1 August 2020) with default parameters to obtain the corresponding annotation file (GTF format). SUPPA2 [33] using default settings generated the AS and transcript events from the GTF file. The AS events generated by SUPPA2 contained five different types: alternative 5’/3’ splice-sites (A5/A3), alternative first/last exons (AF/AL), retained introns (RI), skipping exons (SE), and mutually exclusive exons (MX). For RNA-Seq data, the AS events in all 24 samples were identified based on the GTF file produced by StringTie (v2.1.1) [25]. In order to avoid false positives that might come from short assembly results of RNA-Seq data, we only analyzed the assembly transcripts covering 80% of the gene model or the full-length transcripts.

### 2.11. RT-PCR Validation

RT-PCR was performed to validate the existence of the novel transcripts and AS events by using KOD polymerase (TOYOBO Co., Ltd, Kita-ku, Osaka, Japan). Total RNA was prepared according to the same protocols as for Illumina RNA-Seq library construction and sequencing. cDNA was prepared using a HiScript III 1st Strand cDNA Synthesis Kit (Vazyme Biotech Co., Ltd, Nanjing, China). Specific primers were designed using Primer (v5.0) (Appendix A). The PCR conditions were as follows: a 5 min denaturation at 94 °C followed by 35 cycles of 94 °C for 25 s, 56 °C for 25 s, 72 °C for 5–15 s (depending on different genes), and a final extension at 72 °C for 8 min.

### 2.12. Statistical Analysis and Data Presentation

The statistical analyses and plots were performed using R packages. Heat maps were generated by the pheatmap R package. A Sashimi plot (https://miso.readthedocs.io/en/fastmiso/sashimi.html, accessed on 1 December 2020) was used to show alternative splicing (AS) events and relative abundance. This tool can plot RNA-Seq read densities with exons and junctions as well as visualize the structures of the gene’s isoforms.

## 3. Results

### 3.1. RNA-Seq Data Sequencing and Assembly

To comprehensively understand the molecular information of sex-biased differences during the development of the parasitic wasp *C. vestalis*, cDNA libraries were constructed from male and female *C. vestalis* in different developmental stages (two-day-old~four-day-old pupae and one-day-old adults) using the Illumina HiSeq 2500 sequencing platform. Based on Illumina RNA-Seq, approximately 1083 million raw reads were produced from all samples (Appendix A). After quantity control and filtering, about 1075 million clean reads were obtained, with an average Q30 of 93% (Appendix A). Subsequently, clean reads from each sample were mapped against the *C. vestalis* genome assembly CvesOGS1.0 [20]. Then, corresponding GTF files produced from each sample’s mapping result were merged into a non-redundant GTF file. Finally, a total of 32,102 unigenes were assembled based on the merged GTF file, and 22,196 non-redundant transcripts were obtained using the software CD-HIT (v4.8.1) [27]. The transcripts comprised 13,388 (99.98%) annotated genes of *C. vestalis* and 14,466 novel transcripts (Appendix A).

### 3.2. The Full-Length Sequencing by PacBio Iso-Seq

Total RNA was extracted, and poly(A) RNA was enriched from the 24 mixed samples of pharate adult and adult development of *C. vestalis*. Furthermore, we prepared three full-length cDNA libraries with insert sizes of 1–2, 2–3, and 3–6 kb (Appendix A). Based on PacBio Iso-Seq, 3,837,766 subreads were generated, and we obtained 412,224 circular consensus sequences (CCSs), 207,955 of which were identified as FLNC by the SMRT Portal RS_IsoSeq protocol (Appendix A). The mean lengths of FLNC CCSs were 1471, 2566, and 3482 base pairs (bps) for the 1–2, 2–3, and 3-6 kb insert libraries, respectively (Appendix A). Subsequently, we obtained 78,138 high-quality (HQ) isoforms after error correction using the Illumina RNA-Seq data (Appendix A). Finally, after removing the redundant sequences using CD-HIT (v4.8.1) [27], 76,773 unigenes were obtained, 76,050 (99%) of which were mapped against the *C. vestalis* genome, covering approximately 56.97% of the gene set of CvesOGS1.0 (Appendix A) [20]. Meanwhile, we also identified 11,849 novel gene transcripts using Iso-Seq data (Appendix A). The comparison of length distributions showed that the majority of the novel full-length transcripts were longer than those of mRNAs (Figure 1a). To determine the accuracy and reliability of these novel transcripts, we randomly selected 10 candidates to perform RT-PCR experiments; the results validated all 10 transcripts (Figure 1b).

### 3.3. The Correction of C. vestalis Genome Annotation

Based on the Iso-Seq data, we identified some mis-annotated genes in the *C. vestalis* assembly CvesOGS1.0 [20]. For example, CVE01195 in CvesOGS1.0 was much shorter than the new annotated sequence, as the original annotation of the coding region had some deletions in the first exon and completely missed the fourth exon. Both sequencing results of Iso-Seq and RNA-Seq as well as RT-PCR verified the fourth exon (Figure 1c). By comparing the full-length Iso-Seq transcripts with the reference genome, we found that some annotated genes converged into a single long transcript in Iso-Seq sequencing and were mis-annotated as two separate genes in the reference genome (e.g., CVE02774 and CVE02772, Figure 1c). RT-PCR results verified the newly annotated gene from the full-length transcripts (Figure 1c). RNA-Seq also assembled a transcript linking these two genes in CvesOGS1.0 (Figure 1c). In total, 1592 genes were incorrectly annotated as multiple split genes in the CvesOGS1.0, and these genes could be merged into 650 new genes according to the Iso-Seq data (Appendix A). To determine the accuracy and reliability of these new transcripts, we randomly selected 10 candidates to perform RT-PCR experiments, and the results showed that all 10 were correct (Figure 1d).

### 3.4. LncRNA Identification, Expression Analysis, and Functional Prediction

The Iso-Seq data yielded 8770 lncRNA transcripts located in 331 scaffolds after a stringent filtering process (Appendix A). In order to analyze the structural characteristics of lncRNAs in *C. vestalis,* the length distributions of mRNA transcripts and lncRNAs were compared. Our results showed that almost all of the lncRNAs (99%) were less than 4000 nt in length, and more than half of the lncRNAs were from 1000 to 2000 nt in length, while 29.7% of RNA-Seq assembled mRNAs and 40.1% full-length (FL) mRNAs had the same lengths (Figure 2a–c). As expected, the lncRNA length was the highest among all these RNAs.

The abundance of lncRNAs was quantified and normalized using the FPKM (fragments per kilobase of transcript, per million fragments sequenced) method; the average FPKM values of lncRNAs in each developmental stage are shown in Appendix A. The differential expression of lncRNAs between different developmental times of both sexes in *C. vestalis* (PF2d vs. PM2d, PF3d vs. PM3d, PF4d vs. PM4d, AF1d vs. AM1d, PM3d vs. PM2d, PM4d vs. PM3d, AM1d vs. PM4d, PF3d vs. PF2d, PF4d vs. PF3d, AF1d vs. PF4d) was identified, and the numbers are shown in Figure 2d. The comparison results revealed that a small amount of lncRNAs were significantly expressed between males and females at the same development period (PF2d vs. PM2d, PF3d vs. PM3d, PF4d vs. PM4d, AF1d vs. AM1d), with 100, 16, 62, and 160 lncRNAs up-regulated and 300, 7, 9, and 39 lncRNAs down-regulated, respectively. For the comparison of significant expression differences of lncRNA between different developmental stages, we found over 1400 lncRNAs that were significantly down-regulated between two-day-old pupae and three-day-old pupae of both males and females, while over 1300 lncRNAs were significantly up-regulated between four-day-old pupae and one-day-old adults of both sexes of wasps. The expression profiling results showed that many lncRNAs were specifically expressed in two-day-old male pupae and one-day-old female adults (Figure 2e). Moreover, to detect potential pre-miRNA in lncRNAs, we aligned lncRNAs to a miRbase and previously identified miRNAs in *C. vestalis* [15]. We ultimately found five miRNA precursors located at 24 lncRNA transcripts (Appendix A).

A total of 3717 target mRNA genes of lncRNAs were identified and subjected to GO and KEGG enrichment analyses (Appendix A). Based on the GO enrichment analysis (Appendix A), the most significantly enriched categories in biological process (BP), cellular component (CC), and molecular function (MF) were myofibril assembly, steroid binding, and mitochondrial part, respectively. Based on the KEGG enrichment analysis (Appendix A), the top five pathways were longevity regulating pathway, chemical carcinogenesis, SNARE interactions in vesicular transport, arginine and proline metabolism, and glycosaminoglycan biosynthesis-chondroitin sulfate/dermatan sulfate.

### 3.5. Analysis of Differentially Expressed Genes

In order to comprehensively understand the stage-specific and sex-biased genes in *C. vestalis*, we compared the gene expression levels of pharate adult and adult development both males and females. Differentially expressed genes (DEGs) were filtered based on the log_2_ fold change (log_2_FC > 1) and q-values (*q* < 0.05), and the numbers are shown in Figure 3a. The comparative analyses between male and female wasps at the same development stage (PF2d vs. PM2d, PF3d vs. PM3d, PF4d vs. PM4d, AF1d vs. AM1d) revealed that 1422, 648, 2361, and 3728 genes were up-regulated and 1728, 789, 2091, 2887 genes were down-regulated, respectively (Figure 3a). Meanwhile, the expression levels of over 4000 genes were significantly different when comparing two adjacent development stages of male wasps, whereas the number of DEGs between two adjacent development stages of female wasps varied from >3600 to >6000 (Figure 3a).

Thereafter, stage- and sex-specific analyses were conducted in the eight libraries, resulting in 2125 stage-specific and 326 sex-specific DEGs, i.e., 239 and 87 female- and male-specific expressed genes, respectively (Figure 3b–d, Appendix A). For sex-specific DEGs, we noticed that DEGs in females were separated into two clades: the upper clade contained genes that were specifically expressed in female adults, and the lower clade contained genes that were consistently up-regulated during development in female wasps (Figure 3d). In addition, we found 17 female-specific genes (eight of these proteins had signal peptides) that were highly expressed in one-day-old female adults and in venom glands (Appendix A) that were previously identified as venom proteins [34], including five peptidase family M1/M12A proteins, two *EB module domain proteins*, one *phospholipase A1* gene, one *Chitooligosaccharidolytic beta-N-acetylglucosaminidase* gene, one *Ion transport peptide* gene, and one *Hemocytin* gene. The male-specific genes included four *odorant receptor* genes, four *histone H1* genes, three *leucine-rich repeat-containing protein* genes, and two *alpha tubulin* genes (Appendix A).

To obtain a clearer perspective of stage- and sex-specifically expressed genes, we analyzed the top 10 DEGs in each comparative group (Appendix A). We found no overlap of DEGs present in the comparison of adjacent developmental stages in either male or female wasps. Interestingly, we found two *putative uncharacterized protein* genes, a *glutamine-rich protein 2* gene and a *histone H1* gene, that were persistently highly expressed in male wasps compared with female wasps at the same developmental stage.

### 3.6. GO and KEGG Enrichment of DEGs

The DEGs in different comparative groups were used for the enrichment analyses of Gene Ontology (GO) terms (Appendix A) and Kyoto Encyclopedia of Genes and Genomes (KEGG) pathways (Appendix A). The comparison of adjacent developmental stages of same-sex wasps (PM3d vs. PM2d, PM4d vs. PM3d, AM1d vs. PM4d, PF3d vs. PF2d, PF4d vs. PF3d, AF1d vs. PF4d) revealed that 140, 165, 37, 133, 115, and 36 GO terms were up-regulated, and 83, 53, 93, 9, 46, and 61 GO terms were down-regulated, respectively (Appendix A). Interestingly, we found 13 GO terms (including transmembrane transport, transporter activity, and active transmembrane transporter activity) that were consistently up-regulated with the development of male wasps; three GO terms, i.e., oxidation-reduction process, oxidoreductase activity, and cofactor binding, were persistently up-regulated with the development of female wasps, and three GO terms were enriched in both male and female groups (Appendix A). Furthermore, we found 24 GO terms (such as ncRNA metabolic process, cytoplasmic part, and ligase activity) that were persistently down-regulated in different comparative groups of male wasps, while only six GO terms (such as chitin metabolic process, extracellular region, and chitin binding) were persistently down-regulated in female wasps (Appendix A).

Similarly, analyses revealed significant enrichment of 70, 146, 279, and 376 GO terms in the comparison of different sexes of wasps at the same development stage (PF2d vs. PM2d, PF3d vs. PM3d, PF4d vs. PM4d, AF1d vs. AM1d) with 9, 44, 115, and 205 up-regulated terms and 62, 103, 165, and 172 down-regulated terms, respectively. Obviously, the difference between females and males gradually increased with the development of the wasp. Our results showed that GO terms enriched in female and male wasps were distinct (Appendix A). For example, comparative analyses between AF1d and AM1d indicated GO terms in female adults were significantly enriched in translation (BP), nucleus (CC), and structural constituent of ribosome (MF), while those in male adults were significantly enriched in transmembrane transport (BP), proton-transporting two-sector ATPase complex (CC), and proton-transporting two-sector ATPase complex (MF) (Appendix A).

The comparative analyses of KEGG annotation between adjacent development stages of same-sex wasps (PM3d vs. PM2d, PM4d vs. PM3d, AM1d vs. PM4d, PF3d vs. PF2d, PF4d vs. PF3d, AF1d vs. PF4d) revealed that there were more up-regulated pathways than down-regulated pathways, with 19, 20, 24, 6, 16, and 8 up-regulated and 6, 6, 5, 2, 3, and 7 down-regulated, respectively (Appendix A). Furthermore, we found DEGs up-regulated in the pupal stage were persistently enriched in oxidative phosphorylation, citrate cycle (TCA cycle), carbon metabolism, biosynthesis of amino acids, and phagosome pathways (Appendix A). Comparing pupal stages of the same sex, 15 of 24 up-regulated pathways were specifically enriched in male adult wasps, and 6 of 8 up-regulated pathways were specifically enriched in female adult wasps (Appendix A), and there was no overlap between these two groups.

Meanwhile, KEGG pathway enrichment analysis showed that DEGs between the same development stage of males and females (PF2d vs. PM2d, PF3d vs. PM3d, PF4d vs. PM4d, AF1d vs. AM1d) were enriched in 6, 17, 32, and 55 pathways, with 1, 7, 15, and 16 up-regulated and 5, 10, 27, and 39 down-regulated, respectively (Appendix A). Interestingly, compared with males, we found that the up-regulated DEGs in female wasps from three-day-old pupae were persistently enriched in DNA replication, mismatch repair, ribosome biogenesis in eukaryotes, RNA transport, nucleotide excision repair, and hedgehog signaling pathway-fly (Appendix A), whereas the down-regulated DEGs from three-day-old pupae were persistently enriched in oxidative phosphorylation, citrate cycle (TCA cycle), carbon metabolism, biosynthesis of amino acids, pyruvate metabolism, 2-oxocarboxylic acid metabolism, glycolysis/gluconeogenesis, glycine, serine and threonine metabolism, glyoxylate and dicarboxylate metabolism, and glutathione metabolism (Appendix A).

### 3.7. Identification of Alternative Splicing (AS) Events

In order to identify alternative splicing events in *C. vestalis*, we combined both RNA-Seq and Iso-Seq methods to sequence the pooled samples from different developmental stages and sexes. The genome-wide profiling of AS variants was performed using the long SMRT reads and Illumina short reads using the SUPPA algorithm. In detail, we detected junctions in 76,773 transcripts generated from long SMRT reads after aligning to the *C. vestalis* genome assembly using GMAP [31], and this finally yielded 20,540 splicing isoforms (Appendix A). Meanwhile, a total of 22,791 splicing isoforms were identified in the Illumina transcriptome of 24 *C. vestalis* samples (Appendix A). Overall, 1331 and 4188 genes showing 5043 and 6813 AS events were identified from the Iso-Seq and the RNA-Seq data, respectively (Table 1). Finally, we identified 4819 genes showing 11,856 AS events by combining isoforms from both sequencing methods (Table 1). These isoforms were classified into seven groups (Figure 4a), with 2886 alternative 3’ splice site (A3) (24%), 3040 alternative 5′ splice site (A5) (26%), 2504 alternative first exon (AF) (21%), 475 alternative last exon (AL) (4%), 1512 intron retention (RI) (13%), 1329 exon skipping (SE) (11%), and 130 mutually exclusive exon (MX) (1%) events (Figure 4b, Table 1).

To identify the AS events related to development and sex differentiation of the wasp *C. vestalis*, we compared AS events in different stages of both sexes by combining the Iso-Seq and the RNA-Seq data (Appendix A). Comparing the AS genes among different developmental stages, we found that AS genes were widely (average > 81%) shared in pharate adult and adult development of both sexes. As expected, genes related to 20-hydroxyecdysone (20E) signaling pathways, such as ecdysone 20-monooxygenase (CVE02280) and ecdysone-induced protein 93F (CVE06796), were found to undergo alternative splicing in each stage of both males and females (Appendix A). When comparing males with females at the same stage and time period, the number of AS events in females was slightly greater than in males, except for the two-day-old pupae. Moreover, there was no difference in the distribution of the seven types of AS events between different samples.

### 3.8. Identification of Sex-Determining Genes and Their Transcripts

We identified six key genes related to the sex-determination pathway, i.e., *feminizer* (*Cvfem*, homolog to *D. melanogaster tra*), *doublesex* (*Cvdsx*), *daughterless* (*Cvdau*), *fruitless* (*Cvfru*), *transformer-2* (*Cvtra-2*), and *intersex* (*Cvint*). Furthermore, we found the AS events occurring in these six genes, with 5 (*Cvfem*), 11 (*Cvdsx*), 19 (*Cvdau*), 3 (*Cvfru*), 7 (*Cvtra-2*), and 3 (*Cvint*) splicing isoforms, respectively (Figure 4c–f and Appendix A). Subsequently, we conducted RT-PCR to verify AS events in these genes. The results showed that *Cvfem* had a male-specific exon, and *Cvdsx* had two female-specific exons (Figure 4c–f). The Sashimi plot also clearly showed that *Cvfem* was highly expressed in male pupae, especially in the two-day-old male pupae, and *Cvdsx* had a relatively higher expression level in male wasps compared to female wasps (Figure 4c–f). For the other four genes, two genes (*Cvdau* and *Cvfru*) showed sex-biased exons, with RT-PCR results verifying one female-specific isoform in *Cvdau* and four male-specific isoforms in *Cvfru* (Appendix A). The *Cvtra-2* was mis-annotated as two genes in CvesOGS1.0, while the Iso-Seq full-length reads showed that two isoforms encode a new gene that merged the two mis-annotated genes (Appendix A). We designed a pair of primers located in these two separate genes, and RT-PCR results verified that the two genes were incorrectly annotated and that the two isoforms were correct (Appendix A). The Sashimi plot of *Cvint* showed no sex-biased isoform and no significant difference of expression in different developmental stages (Appendix A).

## 4. Discussion

We combined Iso-Seq and RNA-Seq to construct a comprehensive transcriptome profile of *C. vestalis*, including almost all mRNAs and long non-coding RNAs. The results significantly improved the annotation of the current reference genome of *C. vestalis* CvesOGS1.0 [20] and identified key transcriptomic signatures of major changes in both sexes during the transition from pupa to adult wasp. These new corrected annotations of *C. vestalis* and the comprehensive transcriptome profiling will provide useful information for researching development and sex differentiation in parasitoid wasps.

In recent years, the vast majority of studies on lncRNAs have been reported in mammals [35,36], plants [37], and microorganisms [38]. With the rapid development of sequencing techniques, lncRNAs in many insect species have been identified by RNA-Seq and Iso-Seq in species such as *D. melanogaster* [39], *P. xylostella* [40], *Apis mellifera ligustica* [41], *Nilaparvata lugens* [41], and *Aphis aurantii* [42]. The number of lncRNAs (8770) in this wasp is similar to that of its host *P. xylostella* (8096 lncRNAs), which is more than those of the insects mentioned above [40]. This may be due to the fact that single molecular full-length sequencing could obtain more transcripts than short reads assembly from RNA-Seq. GO enrichment analysis showed target mRNA genes of lncRNAs were strongly associated with many biological processes, such as appendage morphogenesis and development as well as striated muscle cell development, implying that lncRNAs may play a role in wasp pupal and adult development. The development- and sex-associated functions of these specific lncRNAs in *C. vestalis* can be further studied with shorter-interval and sex-specific tissue samples collected for sequencing.

The development of female parasitic wasps usually involves the production of factors related to parasitism, such as venom, polydnavirus, and teratocytes [15]. The DEG analyses identified 17 putative venom proteins that were specifically highly expressed in venom glands and that showed high identity with other wasps in the Hymenoptera (Appendix A). These *C. vestalis* venom proteins included five peptidase family M1/M12A proteins. The homologs in the venom sac of *Pimpla hypochondriaca* showed that these proteins were involved in the processing of peptide precursors [43]. Meanwhile, we identified four *odorant receptor* genes specifically expressed in males that may play important roles in courtship or be involved in the perception of sex pheromones and food source semiochemicals [44,45]. KEGG enrichment analysis showed that genes persistently up-regulated in female wasps were enriched in DNA replication, mismatch repair, ribosome biogenesis in eukaryotes, and nucleotide excision repair. The results are consistent with previous studies reporting that the sizes of cells and their nuclei increase along with DNA content during the development of the wasp ovary, and the cytoplasm of calyx cells is also enriched in ribosomes [46,47]. This sex-specific expression indicates different developmental regulation strategies in female and male wasps.

In this study, we detected 5043 AS events in 1331 genes via Iso-Seq and 6813 events in 4188 genes via RNA-Seq, suggesting that PacBio single-molecule long-read sequencing detected more isoforms for each gene than second-generation sequencing (Table 1). However, fewer genes were shown to produce AS events by Iso-Seq than RNA-Seq, which might be partly due to removal of short fragments (<500 bp) when preparing libraries for Iso-Seq. Similar phenomena have been found in plants and other insects [48,49]. Comparing AS genes among different developmental stages, we found that AS genes are widely (average > 81%) shared in pharate adult and adult development of both sexes, indicating that most genes may have conserved roles in regulation of both male and female development. The results also showed that AS events were more prevalent in females than in males, suggesting that males and females adopt different transcriptional strategies for their developmental regulation.

The genetic sex determination of *C. vestalis* is via haplodiploidy. It is known that the underlying molecular mechanism depends on heterozygosity and homozygosity at multiple complementary sex-determination (CSD) loci [50]. Unlike *A. mellifera*, we found no *csd* in the genome of *C. vestalis*. Thus, we inferred that CSD in *C. vestalis* is regulated by a novel mechanism. Besides the *csd* gene, we identified six key sex-determining genes as well as their alternative splicing isoforms, four genes of which were sex-specifically expressed. In insects, many genes related to sex determination, such as *tra* and *dsx*, have different spliced isoforms that are essential for sex-specific development [51,52,53,54]. AS isoforms of *Cvfem* in male and female wasps are similar to those from *A. mellifera* and *D. melanogaster*, of which male-specific splice variants contain a premature stop codon and yield no functional products, whereas the female-specific splice variants encode the functional proteins [55]. The *Cvfru* gene also showed extremely distinct AS events between males and females, of which male-specific isoforms may lead to male-specific behaviors in response to the same stimulus. For example, male-specific *fruitless* isoforms are involved in determining male courtship behaviors in *Drosophila*, *Musca domestica,* and *Blattella germanica*. [56,57,58,59].

## 5. Conclusions

We combined Iso-Seq and RNA-Seq to perform genome-wide profiling of both sexes of an important parasitic wasp, *C. vestalis*, during pharate adult and adult development. Taking advantage of Iso-Seq full-length reads, we corrected 1592 wrongly annotated genes in the previously obtained gene set CvesOGS1.0 and identified 14,466 novel transcripts as well as 8770 lncRNAs. DEG analyses showed completely different characteristics between males and females, where 2125 stage-specific and 326 sex-specific genes were identified. We also found 4819 genes comprising 11,856 alternative splicing events by combining the Iso-Seq and RNA-Seq results. The results of comparative analyses showed that most genes were alternatively spliced across developmental stages, and AS events were more prevalent in females than in males. Furthermore, we identified six sex-determining genes and verified their sex-specific alternative splicing profiles in this parasitic wasp. The obtained transcriptome will facilitate further studies of genes involved in the regulation of development and sex differentiation in *C. vestalis*.

## Figures and Tables

**Figure 1 genes-12-00896-f001:**
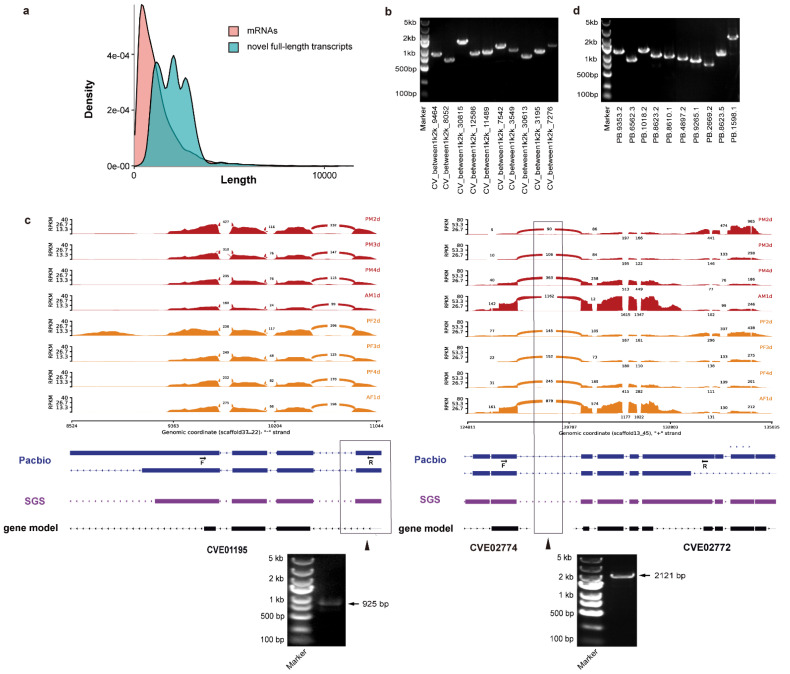
Identification of novel gene transcripts and a revised annotation of the *C. vestalis* genome. (**a**) The comparison of length distributions of novel full-length transcripts and mRNAs in the *C. vestalis* genome. Red: mRNAs, blue: novel full-length transcripts. (**b**) Ten novel transcripts were chosen at random and verified by RT-PCR. (**c**) Sashimi plot of two examples of falsely annotated genes in pharate adult and adult development of both sexes in *C. vestalis*. The thickness of the curves in red (male) and orange (female) crossing two splicing junction sites (SJ) represents the coverage degree of Illumina RNA-Seq reads, and the number of junction reads is marked in the middle of the splicing site. The RNA coverage was given as the log-transformed reads per kilobase of transcript per million mapped reads (RPKM) value. Blocks represent exons, and lines represent introns. The black rectangular box and arrow indicate an extra exon in gene CVE01195 and a junction region in mis-annotated genes (CVE02774 and CVE02772), respectively. Forward and reverse primers used in RT-PCR are marked as F and R, respectively. SGS, second-generation sequencing. (**d**) Ten full-length transcripts that overlapped mis-annotated genes were chosen at random and verified by RT-PCR.

**Figure 2 genes-12-00896-f002:**
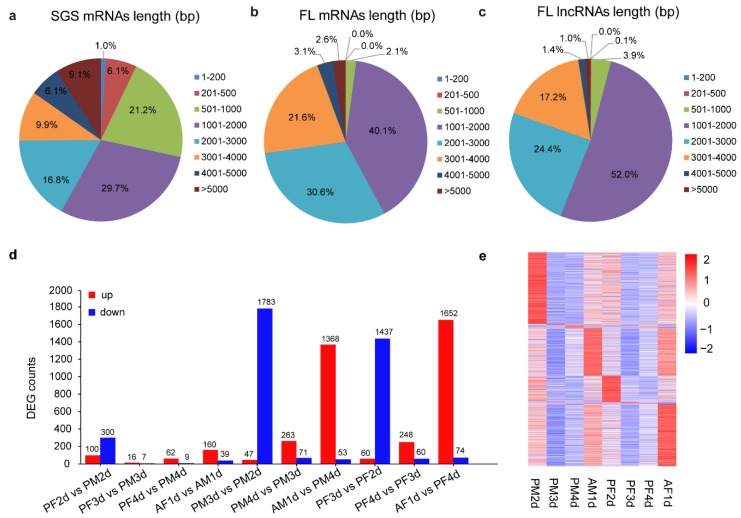
Long non-coding RNA (lncRNA) identification and differential expression patterns of pharate adult and adult development of both sexes in *C. vestalis*. (**a**–**c**) Length distributions of mRNAs identified in SGS data (**a**) and Iso-Seq data (**b**) as well as lncRNAs identified in Iso-Seq data (**c**). FL: full length. (**d**) Analysis of differential expression lncRNAs between pharate adult and adult development of females and males *C. vestalis.* Red bars represent up-regulated and blue bars represent down-regulated lncRNAs. PM2d, PM3d, PM4d, AM1d, PF2d, PF3d, PF4d, and AF1d represent *C. vestalis* two-day-old male pupae, three-day-old male pupae, four-day-old male pupae, one-day-old male adults, two-day-old female pupae, three-day-old female pupae, four-day-old female pupae, and one-day-old female adults, respectively. (**e**) Stage-specific expression patterns of lncRNAs in *C. vestalis.* Red and blue colors in the heat map indicate high and low expression levels, respectively.

**Figure 3 genes-12-00896-f003:**
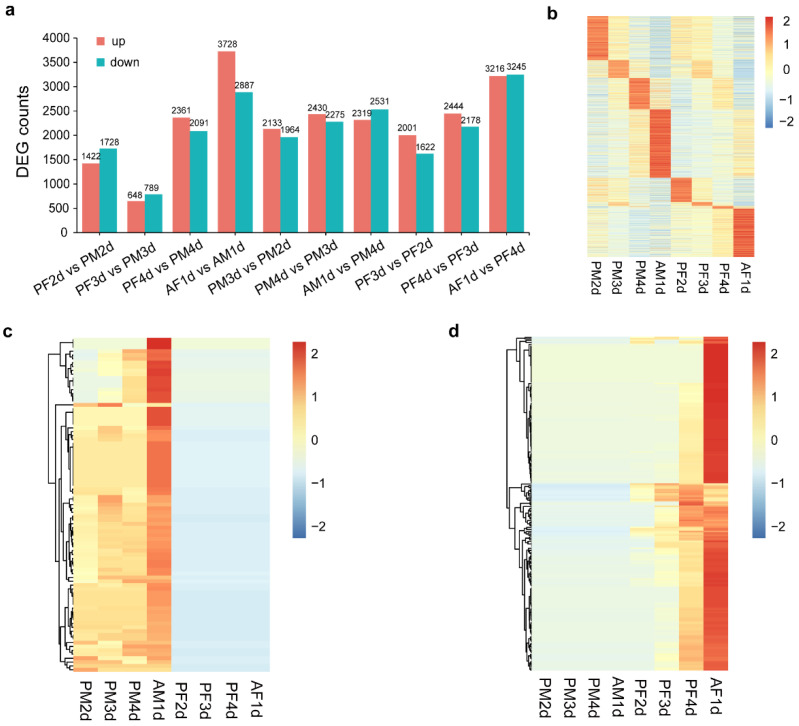
Analysis of differentially expressed genes (DEGs) between male and female wasps of pharate adult and adult development of *C. vestalis.* (**a**) Numbers of DEGs between of pharate adult and adult development of *C. vestalis.* Red bars represent up-regulated genes, and blue bars represent down-regulated genes. (**b**) Stage-specific expression of genes in *C. vestalis.* Red and blue colors in the heat map indicate high and low expression levels, respectively. (**c**) Male-specific genes in all developmental stages. (**d**) Female-specific genes in all development stages.

**Figure 4 genes-12-00896-f004:**
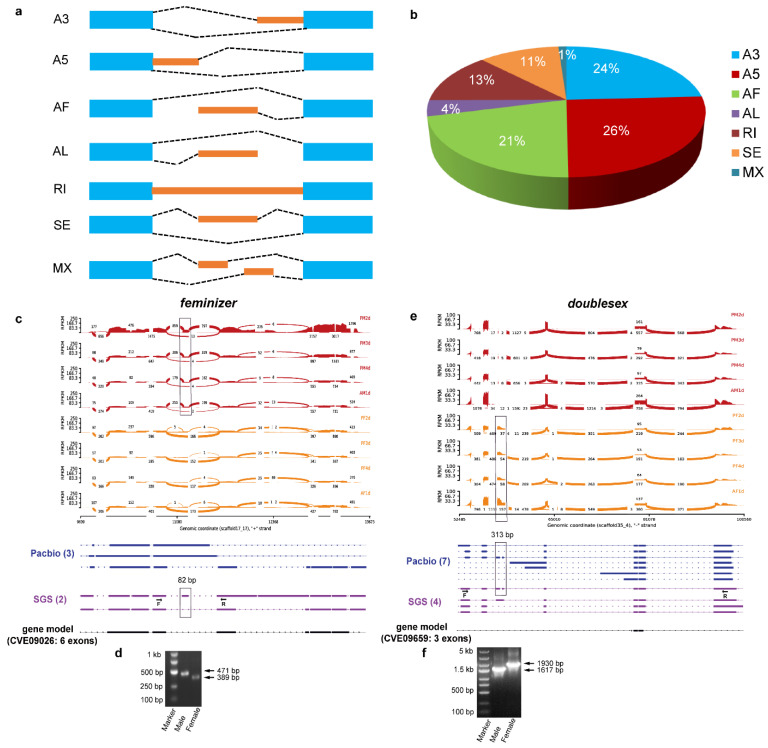
Alternative splicing events analysis in *C. vestalis*. (**a**) Seven types of AS events were identified in *C. vestalis*. (**b**) The proportions of different alternative splicing types identified in *C. vestalis* by combining PacBio and SGS data. (**c**) Sashimi plot of alternative isoforms of *feminizer* in *C. vestalis*. (**d**) A male-specific exon in the *feminizer* gene was verified by RT-PCR. (**e**) Sashimi plot of alternative isoforms of *dsx* in *C. vestalis*. (**f**) Two female-specific exons in *dsx* verified by RT-PCR. The thickness of the curves in red and orange crossing two splicing junction sites (SJ) represents the coverage degree of Illumina RNA-Seq reads, and the number of junction reads is marked in the middle of the splicing site. The RNA coverage is given as the log-transformed reads per kilobase of transcript per million mapped reads (RPKM) value. Blocks represent exons, and lines represent introns. The black rectangular box indicates sex-biased exons selected for RT-PCR verification. Forward and reverse primers used in RT-PCR were marked as F and R, respectively. SGS, second-generation sequencing.

**Table 1 genes-12-00896-t001:** Alternative splicing events in splicing isoforms identified in Iso-Seq and RNA-Seq data by SUPPA.

	PacBio		SGS		Combination	
Type	Genes	Events	Genes	Events	Genes	Events
A3	412	644 (13%)	1803	2222 (33%)	2092	2866 (24%)
A5	496	793 (16%)	1874	2247 (33%)	2265	3040 (26%)
AF	327	1889 (38%)	472	615 (9%)	792	2504 (21%)
AL	63	289 (6%)	140	186 (3%)	194	475 (4%)
RI	542	930 (18%)	523	582 (9%)	1065	1512 (13%)
SE	296	425 (8%)	781	904 (13%)	973	1329 (11%)
MX	45	73 (1%)	52	57 (1%)	97	130 (1%)
Total	1331	5043	4188	6813	4819	11,856

SGS: second-generation sequencing.

## Data Availability

The raw sequence reads of RNA-Seq and Iso-Seq are available in the NCBI BioProject repositories with the accession numbers PRJNA701718 and PRJNA701114, respectively.

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
