# Peer review of "Comparative Transcriptome Analysis Reveals Sex-Based Differences during the Development of the Adult Parasitic Wasp *Cotesia vestalis* (Hymenoptera: Braconidae)"

_genes, 2021, doi:10.3390/genes12060896_

Round 1

Reviewer 1 Report

This study delves into the genetic difference between sexes of the parasitic wasp Cotesia vestalis, and clearly represents a lot of work, with parallel analyses at multiple points in development in both sexes. It even provides important revisions to the existing C. vestalis genome, using long seq data. The paper itself seems actually to be of two minds as to its aims. First, to find which genes are differentially expressed in males and females and thus may be responsible for behavioral differences (e.g. odorant receptors in males). Second, to interrogate differences in genomic structure and function which may contribute to sex determination (e.g. alternative splicing in specific loci identified in other taxa such as transformer). It ultimately lays important groundwork and will be essential reading for anyone working with C. vestalis genomics in the future.

General comments

My major critique is that this paper is of two minds with its aims but is not organized as such. First, I think the introduction needs to be revised to show that two aims will be addressed with the same data, but that these are separate questions. As the paper is currently, the first paragraph leads me to expect a paper that is about genome structure/function and splicing. Then the second paragraph, which is more on the insect itself and talking about how sex differences can serve a practical function in biocontrol, completely threw me off. In the introduction, the authors should clearly lay out their different aims, then follow that structure throughout. Authors should also try to integrate these two aims somehow within the discussion.

The abstract also could be a little more focused. For instance, it reports gene numbers that are fairly meaningless with no context, rather than trends and why they’re interesting, or why lncRNA is interesting in regards to sexual differentiation.

More specific comments

  • The introduction needs more background information on haplodiploidy and what that means for sex determination and sex differences. Rather than the second paragraph using the status of the parasitoid as a biocontrol tool against a pest species, why not motivate this study of genomic mechanisms of sexual development with facts about hymenopterans? E.G. that hymenopterans are haplodiploid and that automatically changes how sex determination works on a genomic level versus the citations given in drosophila.

For instance, see these citations:

Heimpel, George E., and Jetske G. De Boer. "Sex determination in the Hymenoptera." Annu. Rev. Entomol. 53 (2008): 209-230.

Beukeboom, Leo W., and Louis Van De Zande. "Genetics of sex determination in the haplodiploid wasp Nasonia vitripennis (Hymenoptera: Chalcidoidea)." Journal of genetics 89.3 (2010): 333-339.

  • In line 174, the authors state that lncRNAs were not functionally annotated in other databases, but were there orthologues in related species, supporting their function via their evolutionary conservation? The authors assume gene function by looking for upstream genes and going from their GO terms, but homology can also be seen in the lncRNAs themselves, across taxa. See this citation:

Noviello, Teresa MR, Antonella Di Liddo, Giovanna M. Ventola, Antonietta Spagnuolo, Salvatore D’Aniello, Michele Ceccarelli, and Luigi Cerulo. "Detection of long non–coding RNA homology, a comparative study on alignment and alignment–free metrics." BMC bioinformatics 19, no. 1 (2018): 1-12.

  • On lines 183-184, authors should explain the parameters by which read count was normalized within and between individuals.
  • The graphs and heat maps in figures 3 and 4 could be recolored so they don’t look the same, as they’re measuring differently expressed lncRNAs and then just basic differentially expressed genes between stages, and this is not really clear just be looking at figures that essentially look the same.
  • On line 410 – GO term enrichment increased gradually in both sexes – was it faster in males than females as males eclosed sooner? A figure of this showing the rates of enrichment between the sexes, perhaps in the supplement, would work to show this.
  • In Table 1, how is it that the different sequencing techniques did not identify the same genes. In other words – why does PacBio + SGS numbers of genes = approximately the combined number, rather than a larger portion of the genes found using different techniques being duplicates? If each technique finds unique splicing isoforms, how can any count of them be compared?

Reviewer 2 Report

This reviewer found nothing out of order with the methods, data analysis, presentation of results and discussion of findings.  The manuscript seems to be a well done standard survey of genes expressed in male and female parasitoids.  Here are specific comments and editorial suggestions, which can be passed on to the Authors’.

Line 3, title:  Change “development of the parasitic wasp” to “development of the adult parasitic wasp” to indicate the focus of the study on that stage.

Line 19:  Change “profiling of male and female C. vestalis at multiple developmental stages” to “profiling of pharate adult and adult development of male and female C. vestalis” to indicate specifically what parts of adult development were examined.  Make appropriate adjustments using this terminology in all other parts of the manuscript.  

Lines 27-290:  Delete these sentences which say absolutely nothing except the obvious about what you found.  You could replace them with something specific about what was found about regulation of transcription and splicing, which was not stated in the sentence in lines 25-26.

Figure 1:  Move to supplemental data as highlights of the bioinformatic processes are given in the previous and following paragraphs.

Line 13:  Font used in this line is different from everything else.

Results section 3.4 on lncRNAs findings:  Indicate if the C. vestalis genome was updated with annotation of these sequences like what was done for mis-annotated genes.
